# The Individual Nutrition Education Needs among Patients with Type 2 Diabetes at the Public Health Centers in Padang, Indonesia: A Cross-Sectional Study

**DOI:** 10.3390/nu14051105

**Published:** 2022-03-05

**Authors:** Ice Yolanda Puri, Barakatun-Nisak Mohd Yusof, Zalina Abu Zaid, Amin Ismail, Hasnah Haron, Nur Indrawaty Lipoeto

**Affiliations:** 1Department of Dietetics, Faculty of Medicine and Health Sciences, Universiti Putra Malaysia (UPM), Serdang 43400, Malaysia; iceyolandapuri@ph.unand.ac.id (I.Y.P.); zalina@upm.edu.my (Z.A.Z.); 2Department of Nutrition, Faculty of Public Health, Andalas University, Padang 25175, Indonesia; 3Institute for Social Sciences Studies, Universiti Putra Malaysia, Serdang 43400, Malaysia; 4Department of Nutrition, Faculty of Medicine and Health Sciences, Universiti Putra Malaysia (UPM), Serdang 43400, Malaysia; aminis@upm.edu.my; 5Nutritional Science Programme, Centre of Healthy Ageing and Wellness, Faculty of Health Sciences, Universiti Kebangsaan Malaysia, Kuala Lumpur 50300, Malaysia; hasnaharon@ukm.edu.my; 6Department of Nutrition Sciences, Faculty of Medicine, Andalas University, Padang 25127, Indonesia; indralipoeto@med.unand.ac.id

**Keywords:** nutritional practices at PHC, patients’ with type 2 diabetes perspective, individual nutrition education

## Abstract

*Background*: The Indonesian Public Health Care (PHC) of Management Nutrition Therapy (MNT) guidelines describe that individual nutrition education is aimed to improve quality of MNT services. The guidelines were originally developed for non-communicable diseases (NCDs), not specially for type 2 diabetes mellitus (T2DM) purposes. The reluctance of patients with T2DM to attend individual nutrition education is a common public health care issue in Padang (Indonesia). *Methods*: The presented cross-sectional study aimed to determine the individual nutrition education needs among people with T2DM. A set questionnaire was distributed to 11 PHC selected from 11 districts in Padang and 179 patients with T2DM were recruited and interviewed. *Results*: Among the 179 patients with T2DM, 76.5% were females and housewives (49.2%), a slight majority (57.8%) were ≤58 years old and 45.9% had graduated from primary school. The highest numbers of patients with T2DM were in PHC Andalas (20.7%). Some 74.9% (134) of the people with T2DM routinely attended individual nutrition education classes for less than 30 min (60.3%). Patients with T2DM followed individual nutrition education at a PHC every 1–2 months (59.8%), and a majority of the individual nutrition education was given by a medical doctor (57.5%). In contrast, 42.3% of patients with T2DM did not attend individual nutrition education due to a lack of recommendation from a medical doctor and their reluctance (33.3%). Although a majority of patients with T2DM (62.6%) were satisfied with their individual nutrition education, (20.4% of patients with T2DM recommended the availability of booklets during individual nutrition education that can be read at home. Patients with T2DM needed individual nutrition education (88.8%) and the majority (25.1%) requested individual nutrition education topics about diabetes food recommendation. Even though patients with T2DM followed health professionals’ advice (78.2%), however, their HbA1c (76.5%) wasnot reduced. Patients with T2DM agreed that individual nutrition education can increase their knowledge (51.9%), unfortunately, they still have difficulty to control their blood glucose (5.6%). *Conclusions*: According to the patients with the T2DM perspective stated above, it is crucial to develop the tool kits and educate patients with T2DM following the Diabetes Nutrition Education (DNE) curriculum to improve glycemic control.

## 1. Introduction

Type 2 diabetes mellitus (T2DM) occurs when the blood glucose level rises excessively because the ineffective body use of insulin. The International Diabetes Federation has estimated that diabetes prevalence has increased by 51% globally, including about 74% in South-East Asia [1]. Indonesia is no exception to this trend. The prevalence has increased steadily from 7% in 2016 to 10.9% in 2018 [2]. Padang, the capital city in West Sumatera, has not observed the same rises. DM prevalence there increased slowly from 1.3% in 2013 to 1.8% in 2018 [2].

Nutrition education, which can be presented in an individual or group setting, is an essential component in T2DM management to learn appropriate nutrition skills. Nutrition education is a method to educate, develop, and maintain the behaviour changes made by patients by enhancing good eating habits [3]. One of the essential components in nutrition education includes meal planning guided by the change behavior model. The purpose is to support the process, set priorities, establish goals, and improve the quality of life [4]. 

The critical needs of nutrition-related knowledge in facilitating lifestyle changes are well established [5,6,7]. Consumption of healthy diet is a key to improving T2DM control. Medical nutrition therapy (MNT) guidelines recommend a reduction in energy and saturated fat intake but increased consumption of high fiber foods (fruits and vegetables), whole grains, and legumes [5,6,7]. Furthermore, patients with T2DM require a set of skills in managing their carbohydrate intake, which can be thought to include carbohydrate counting, exchange systems, food pyramids, healthy plates and meal planning depending on their skills and comprehension. Others proposed topics include explaining diabetes pathophysiology, meal frequency, cooking methods, fiber and whole grains intake and healthy eating [7,8,9,10]. In The Netherlands, when patients received quality education with face-to-face consultations, they did not benefit much from 3–6 months visits. Patients still embarked on unhealthy lifestyles practices, which suggests the critical need for self-management, especially lifestyle changes [11]. Hence, nutritionists released a community-based food education program that significantly increased nutrition-related knowledge provided via face-to-face. This is because face-to-face education allows individuals with T2DM to ask questions and discuss issues [12]. 

We previously observed that the nutritionists perceived that patients with T2DM were reluctant to attend nutrition education sessions because the education materials were not interesting [13]. Moreover, nutritionists have also perceived that the patients felt fewer benefits from individual nutrition education [13,14]. Therefore, a nutritionist should consider patients’ needs when developing nutrition education materials, which is crucial to delivering nutrition information related to diabetes. Nonetheless, knowledge about practices and perception, as well as our patients’ needs, is scarce. Therefore, this study aimed to determine the need for nutrition education among individuals with T2DM. The results would inform the decision in producing individual nutrition education for patients with T2DM at PHCs in Padang (Indonesia).

## 2. Material and Methods

### 2.1. Study Design

A cross-sectional survey was conducted among patients with T2DM at selected PHCs. A 30-item questionnaire survey that included multiple-choice and open-ended questions was used to allow subjects to deliver their responses in their own way [15]. The patients with T2DM were interviewed about sociodemographics, the individual nutrition education for T2DM patients at PHC, and the need for individual nutrition education. Selected and trained enumerators and who have a nutrition education background delivered the questionnaires, which took around 20 min. The Faculty of Medicine, Andalas University (No. 423/KEP/FK/2019) and the Ethical Committee of Universiti Putra Malaysia (JKEUPM-2019-423) approved the study in 2019. 

### 2.2. Respondents, Sampling and Study Location

Respondents (both male and female) who participated in the survey had a confirmed diagnosis of T2DM for at least 6 months, were aged between 18 and 65 years old, and could read, write and communicate well. Respondents were excluded from the study if they had severe diabetes complications such as stroke and renal failure, were pregnant or breastfeeding. 

The formula used to calculate the sample size used the cross-sectional study by Aday and Cornelius et al. [16] and the standard deviation (HbA1c), also following a previous study (Rusdiana et al., [17]), as follows:$$n=\frac{Z^{2}\left(1-\frac{\alpha }{2}\right)\sigma^{2}}{d^{2}}$$
where, σ = estimated standard deviation = 1.8 [17], d = desired precision (1), Z^2^_1−__α_ = 0.95 = 1.96. A total of 179 patients were recruited based on PHC visits from June to July 2020.

### 2.3. Study Site and Sampling Procedure

The study was conducted at Public Health Clinics (PHCs, known as PUSKESMAS in the Indonesian language), located in Padang (Indonesia). These PHCs were located in 11 districts selected randomly from among the 23 available PHCs. One PHC was selected for each district (Figure 1). 

Eligible patients based on the criteria were selected using simple random sampling at the 11 selected PHCs. Therefore, the 179 subjects are divided into 11 districts, with 16 or 17 T2DM patients from each PHC. 

### 2.4. Questionnaires 

The questionnaire consisted of 32 questions in the form of multiple-choice (24 items) and open-ended (six items) questions. The questions were developed based on the Integrated Health Service and Promotion questions for the prevention of non-communicable diseases [15]. The questions were then modified to identify the need for the current nutrition practice among patients with T2DM [11]. This questionnaire consisted of three sections divided into sociodemographics (eight questions), nutrition practices and perception of individual nutrition education received at PHCs (14 questions), and the need for individual nutrition education (10 questions).

Enumerators with nutrition education backgrounds were trained to interview the respondents. The interview duration for each patient was less than 20 min. Especially in the case of elderly patients, some of them visited the PHC alone and the rest were accompanied by family members. The questionnaire was pre-tested among 20 patients with T2DM recruited from PHCs in districts outside Padang.

### 2.5. Data Analyses

Descriptive statistics were reported as frequencies and analyzed in SPSS version 25 (SPSS, Chicago, IL, USA). The descriptive statistics were used to describe the respondent’s information, including sociodemographic characteristics, current nutrition practices, and the need having individual nutrition education

## 3. Results

A total of 179 respondents participated in the study, of which 76.5% were females, housewives (49.2%), with an average age of 57.5 years old and had attained at least a primary school level education (45.9%). The highest numbers of respondents were from PHC Andalas (20.7%, Table 1).

Table 2 shows that 74.9% of respondents routinely attended an individual nutrition education session for less than 30 min (60.3%) every 1–2 months (59.8%), that was mainly provided by a medical doctor (57.5%). Most respondents were satisfied with the education sessions (65.4%) and perceived they had enhanced their overall understanding (73.2%) of nutrition. The most common nutrition education materials used by providers were leaflets (46.9%) and food models (24.0%), with the most common topics being covered foods high in sugars (28.5%), the pathophysiology of diabetes (26.8%), signs and symptoms (26.3%) and sugar consumption recommendations (22.9%). Overall, most respondents were satisfied with the current education materials (62.6%), but 20.4% recommended that booklets be provided to read at home. 

About 25.1% of respondents who did not attend the visit routinely expressed that their doctor had not referred them to a nutritionist (42.3%) or they not interested in the sessions (33.3%). More than 10% perceived that individual nutrition education sessions would only be needed once upon diagnosis, and a small percentage of the respondents perceived their diabetes was under reasonable control (2.2%) (Table 2).

**Table 2 nutrients-14-01105-t002:** Nutrition practices and perception about individual nutrition education received at Public Health Center (*n* = 179).

Nutritionists Practices Component	*n*	%
Attendance		
- Routine visit - Not routine	134 45	74.9 25.1
Frequency		
- Every 1–2 months - Every 3–6 months - Every >6 months	107 7 20	59.8 3.9 11.1
Duration		
- <30 min - ≥30 min	108 26	60.3 14.5
Main provider		
- Medical Doctor - Nutritionists - Nurse	103 89 20	57.5 49.7 11.2
General understanding towards education sessions		
- Good - Moderate	131 3	73.2 1.7
Satisfaction of education session		
- Satisfied - Not satisfied	117 18	65.4 10.1
Nutrition education materials		
- Leaflet - Food Model - No tool kits - Poster - PowerPoint slides	84 43 37 14 4	46.9 24.0 21.0 7.8 2.3
Satisfaction towards the tool kits		
- Satisfied - Not satisfied	112 22	62.6 12.3
Recommended toolkits		
- Booklet - Leaflet - Food card - Flipchart	36 26 18 11	20.4 14.2 10.1 6.1
Coverage of nutrition education topics		
- Food high in sugars - Pathophysiology of diabetes - Sign and symptoms - Sugar recommendations - Cooking process - Meal plan - Food-based recommendation - Exercises - Fiber recommendations - Plate method - Macronutrient recommendations - Nutritional status - Carbohydrate counting - Food models - Carbohydrate exchange - Balanced nutrition guidelines - Food labelling - Complications of diabetes - Sleep patterns - Ways to reduce fasting blood glucose	51 48 47 41 37 37 33 25 19 17 17 14 9 8 6 6 4 2 1 1	28.5 26.8 26.3 22.9 20.7 20.7 19.8 14.0 10.6 9.5 9.5 7.8 5.0 4.5 3.4 3.4 2.2 1.2 0.6 0.6
Common nutrition assessments		
- Fasting blood glucose - Medications - Body weight and BMI - Diabetes nutrition education - HbA1c - Waist circumference	131 114 116 103 64 55	73.2 63.7 64.8 57.5 35.8 30.7
Perception of patients who did not routinely attend individual nutrition education		
- Not recommended by a medical doctor - Not interested - Only once when were diagnosed with T2DM - Focus on diabetes medications - Perceived good control - Perceived needed when not in good control	19 15 6 3 1 1	42.3 33.3 13.3 6.7 2.2 2.2

Most of the respondents (88.8%) expressed a need for nutrition education (Table 3). About 66% had actively asked questions and the questions were mainly about diabetes foods (25%), and other questions such as strategies to reduce blood glucose (5.6%) and no improvement was observed even when patients adhered to education (x%). Although the majority of respondents (78.2%) perceived that they followed the education given (78.2%), their improvement in blood glucose was minimal (23.5%). For those who were unable to follow the education given (17.3%), their reasons included reluctance to follow the instructions, lack of family support (%) and special events such as Eid Mubarak (Table 3). Respondents agreed on the benefits of applying the nutrition component, which includes knowledge increase (51.9%), quality of life increase (27.9%), blood glucose decline (10.0%), and body weight decrease (1.1%).

About 25.1% of patients with T2DM suggested as individual nutrition education topics several menu plans, 11.2% symptoms and etiology, and 9.5% the pathophysiology of diabetes. Therefore, booklets were perceived as appropriate toolkit for reference by patients with T2DM at home.

**Table 3 nutrients-14-01105-t003:** The need for diabetes nutrition education among T2DM patients at PHCs in Padang (*n* = 179).

Items	*n*	%
The need for individual nutrition education		
- Needed - No need	159 19	88.8 10.6
Perception that individual nutrition education is not needed (*n* = 179)		
- Boring - Reluctant - Complicated	12 4 2	7.2 2.3 1.1
Common questions asked during education sessions		
- About diabetes food recommendations - Strategies to reduce blood glucose - Reasons for no improvement despite following the advice - Usage of medication - Recovery from diabetes - Management of diabetes symptoms - Portions of foods - Complications of diabetes - Weight management - Sugar recommendations - Other questions	45 10 10 9 9 9 7 4 2 2 12	25.1 5.6 5.6 5.0 5.0 5.0 3.9 2.3 1.1 1.1 7.2
Respondents who actively participated during an education session		
- Asked questions - No question	118 61	66.0 34.0
Able to follow the education		
- Yes - Unable	148 31	82.7 17.3
Perception of reasons for not following the education		
- Reluctance - Family member influence when they bring food - Hunger - Parties for Eid Mubarak - Nutritionists are rushing because the number of patients is excessive	20 4 2 1 1	11.1 2.3 1.1 0.6 0.6
Recommended topics		
- Menu plans - Symptoms and etiology - Pathophysiology of diabetes - Carbohydrate exchange - High sugar foods - Balanced Nutrition Guide (*Pedoman Gizi Seimbang*) - Plating methods - Provide prescriptions on sugar consumption - Provide macronutrients - Carbohydrate counting	45 20 23 1 1 11 2 1 7 7	25.1 11.2 9.5 0.6 0.6 6.1 1.1 0.6 3.9 3.9
Perceived improvement in glucose control		
- Not improved - Improved	137 42	76.5 23.5
Perceived benefits of education		
- Knowledge increase - Quality of life increase - Blood glucose decline - Body weight decrease	93 50 18 2	51.9 27.9 10.0 1.1

## 4. Discussion

The study examined the perspectives of patients with T2DM about diabetes nutrition practice adequacy at PHCs in Padang (Indonesia). Patients with T2DM were recruited: (1) when they were diagnosed as new DM patients, (2) had uncontrolled fasting blood glucose, (3) were Prolanis members (T2DM patients who are registered with the Indonesian insurance program). A majority of the T2DM patients were female (76.5%), 45.9% of them had graduated from primary school and the average age was 57.51 years old. In Indonesia, individual nutrition education should follow management nutrition practices at PHCs established by the Ministry of Health. T2DM patients should receive nutrition counseling besides anthropometry measurements, biomedical and clinical measurements. In Portugal, the primary health care system with community-based intervention provides food education programs and exercise programs as nutrition intervention to improve behavioral change. Nutrition educators used face-to-face education that allowed T2DM patients to ask questions and discuss concerns. Unfortunately, food education programs report lower attendance compared with other educational interventions in T2DM patients [18]. Diabetes care in The Netherlands is currently focused on regular face-to-face consultations for self-management activities such as medical management (DM medicine and dietary advice), behavior and emotional management. Although T2DM patients had adequate HbA1c (≤60 mmol/mol), however, T2DM patients stated that they perceived no great benefit from the evidence-based guidelines for treatment of T2DM [11]. 

The study found that T2DM patients stated that there was often no recommendation of a medical doctor (42.3%) for education, and they were also reluctant to attend nutrition counseling (33.3%). Although T2DM patients have good perceptions of nutrition education (73.2%), and were mostly satisfied with the educational aid (62.6%), T2DM patients suggested providing booklets (20.4%) during nutrition counseling. A current study of the perspective of patients with T2DM in Mexico reported that books and infographics were the most common frequently used tools in individual nutrition education and resuted in improved HbA1c levels [19]. Individual nutrition education improved HbA1c if patients with T2DM had good diabetes nutrition knowledge and awareness of nutrition education [4]. Individual nutrition education played a role key in diabetes nutrition services and the DM management diet of each individual with T2DM [19]. It is crucial to increase the attendance of patients with T2DM at PHC educational sessions. Several barriers should be considered to avoid bias. These barriers were misunderstanding of the effectiveness of individual nutrition education, confusion regarding when, and how to make referrals, lack of access nutrition services, and the psychosocial and behavioral factors of patients with T2DM [20], including advanced age, and memory loss [19], education level, duration having of DM, and baseline knowledge of DM [6]. Additionally, a cross-sectional study in Saudi Arabia and China mentioned that transportation difficulties, forgetfulness and being busy had an influence on keeping regular medical doctor appointments, regularly attending clinics [20], while lack of interest, and being unable to be contacted, respectively, were mentioned as common barriers in patients with T2DM [21]. Higher attendance of individual nutrition education may impact by T2DM prevalence by increasing knowledge levels. As common education methods, face-to-face nutrition education allowed patients with T2DM and nutritionists to ask questions and discuss issues [18]. To improve glycemic control, patients with T2DM should have good DM knowledge and follow a DM diet and nutritionists should prepare and deliver active and effective dietary education classes [22]. 

A majority of T2DM patients (66.0%) asked question during nutrition counseling follow the advice of the health professional (82.7%), and their knowledge increased (51.9%). Unfortunately, T2DM patients are still reluctant to attend nutrition education classes (11.1%), and perceived no improvement in HbA1c levels (76.5%). A study about MNT from an Indian perspective indicated the importance of developing individual nutrition education topics and providing meal plans with a focus on nutrition intake, healthy food choices, good eating patterns, and food portion size [23]. T2DM patient’s perspective in The Netherlands indicated they needed to increase their DM knowledge, to adhere to DM diet management practices, exercise, and regular schedules. Therefore, it is necessary to consider developing tools and strategies, and assess what T2DM patients’ need are [11]. 

In this study, T2DM patients mentioned that the most common nutrition education topics that they received are education about high sugar food (28.5%), the definition of DM (26.8%), and symptoms and etiology (26.3%), respectively. The most common question are about DM food recommendations (25.1%), compared to how to reduce blood glucose (5.6%), and why blood glucose does not decrease even after following health professionals’ advice (5.6%). Also, T2DM patients had other question during nutrition counseling such as DM menus, DM meal times, DM exercises and vegetables. Therefore, 25.1% of T2DM patients requested DM menu plans, and symptoms and etiology of DM (11.2%) as nutrition education topics. Since the majority of subjects are elderly from the nutritionists’ perspective, patients with T2DM find it difficult to remember the lectures on individual nutrition education topics, therefore, it is crucial to provide appropriate toolkits. According to diabetes health education in Portugal and China, and the perspective of patients with T2DM in Indonesia, booklets are appropriate to deliver individual nutrition education in Indonesia. Patients with T2DM are not familiar with the use of Android for online individual nutrition education. The booklet can address the needs of all patients with T2DM questions during individual nutrition education and practices in their house. 

Similar to many of the abovementioned countries, Indonesia also faces challenges in providing individual nutrition education to patients with T2DM. In this study, among those who did not receive education, 42.3% of them claimed it was not recommended by their doctors, while another 33.3% were reluctant to attend individual nutrition education classes. Another local study among people with T2DM found that a high level of carbohydrate intake was significantly associated with poor HbA1c. Those who did not adhere to the diet recommendations also had significantly poorer HbA1c status [18]. 

Very often, elderly T2DM patients face more difficulties in remembering the lecture content at individual nutrition education classes, therefore, toolkits should be provided to enhance the learning experience. Studies in Portugal, China, and Indonesia have proven that booklets are effective at delivering individual nutrition education to patients with T2DM, especially those who do not have access to or are not familiar with online individual nutrition education. The booklets can be used to address any T2DM questions from the patients that may arise during individual nutrition education and home visits.

## 5. Conclusions

Globally, many countries face challenges in providing effective individual nutrition education to improve the health status of diabetic patients. Indonesia and other countries have similar problems in diabetes individual nutrition education. With the prevalence of T2DM increasing every year, it is crucial to consider a strategy to improve glycemic control in patients with T2DM. Individual nutrition education can offer a diabetes-specific curriculum to educate patients with T2DM and provide knowledge for improving their quality of life. 

## Figures and Tables

**Figure 1 nutrients-14-01105-f001:**
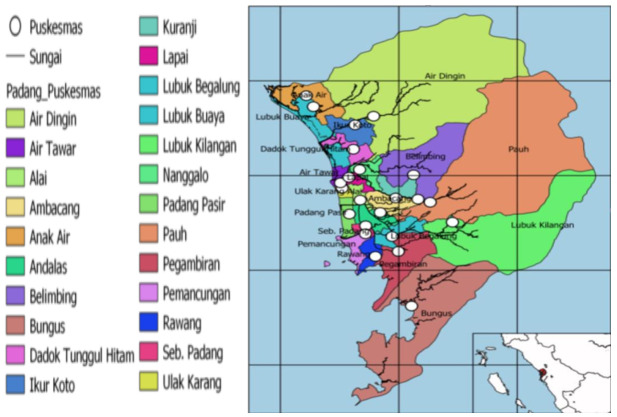
Public health care (PHC) location map of Padang (Google).

**Table 1 nutrients-14-01105-t001:** Sociodemographic characteristic of patients with T2DM at selected PHCs in Padang (*n* = 179).

Items	*n*	%
Sex		
- Males - Females	42 137	23.5 76.5
Age, years		
Mean SD	(57.51 ± 9.61)	
- ≤58 - >58	103 76	57.8 42.5
Educational level		
- Primary school - Secondary school - Tertiary school	82 74 23	45.9 41.3 12.8
Employment		
- Housewife - Currently unemployed - Currently employed	88 46 45	49.2 25.7 25.1
Participating Public Health Care (PUSKESMAS)		
- Andalas - Nanggalo - Air Tawar - Lubuk Begalung - Belimbing - Pauh - Rawang - Padang Pasir - Air Dingin - Lubuk Kilangan - Bungus	37 21 18 17 17 16 14 12 11 10 6	20.7 11.7 10.1 9.5 9.5 8.9 7.8 6.7 6.1 5.6 3.4

## Data Availability

No supplement data.

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
