# Peer review of "The Individual Nutrition Education Needs among Patients with Type 2 Diabetes at the Public Health Centers in Padang, Indonesia: A Cross-Sectional Study"

_nutrients, 2022, doi:10.3390/nu14051105_

Round 1

Reviewer 1 Report

Dear Authors,

This manuscript provides an important resource for understanding the reality of individual nutrition education among patients with Type 2 Diabetes Mellitus (T2DM) and suggests issues that need to be improved in order to increase the effectiveness of nutrition education for T2DM.

Several points as indicated below should be addressed by authors to improve the quality of the article, however.

  1. It should be stated in the “Material and Methods” section how the interview with the T2DM patient was conducted. Was the interview conducted by doctors, nurses, nutritionist, or other research staff with no specific qualifications? How long was the interview conducted for each patient? In the case of elderly patients, was the interview conducted in a situation where they were alone, or were other family members sometimes present?

  1. page 6, line 137-139

The authors describe that “About 14.5% of individuals with T2DM delivered for individual nutrition education topics several DM food recommendations, 10.6% all about DM, and 7.3% management DM diet.” But the figures 14.5% and 7.3% are not in Table 3 and I can’t understand what they mean by above description. It should be reworded to make it easier to understand.

  1. Why did the authors use “58 years old” for stratification of patients by age? 60 or 65 years old may be usually used for such stratification.

  1. In the “Discussion” section, the authors should address limitation of their study. The number of patients, 179, is not necessarily large. No analyses is conducted for association between items in “Sociodemographic characteristics” (Table 1) and those in “Current practice“ (Table 2) or between items in “Sociodemographic characteristics” (Table 1) and those in “The need“ (Table 3). Result from such association analyses may improve the quality of their work, which is expected in future studies.

Author Response

Nutrients

MDPI, St. Anlage 66

CH-4052 Basel, Switzerland      

nutrients@mdpi.com

Telp:+41 61 683 77 34

 Fax: +41 61 302 89 18

Dear Editor of Nutrients, 

We would like to express tremendous gratitude for the email on 5th February 2022, and the opportunity to revise of manuscript ID- 1585719. We also thank the valuable comments from the reviewers that improved this final manuscript. 

The manuscript is revised based on the reviewer comments accordingly. We very much hope the revised manuscript is accepted for publication in Nutrients.

Yours sincerely

Ice Yolanda Puri

Table confirmation of proofreading Nutrients ID-1234331 (T2DM Patients Perspective)

  1. Reviewer-1

No

Item

Correction

Comments

Confirmation

Page

1

Material and Methods

It should be started in the “Material and Methods” section how the interview conducted by doctors, nurses, nutritionists, or other research staff with no specific qualifications? How long was the interview conducted for each patient? In the case of elderly patients, was the interview conducted in a situation where they were alone, or were other family members sometimes present?

As mentioned on the material and methods

3

2

Results

Table 3

The authors describe that “About 14.5% of individuals with T2DM delivered for individual nutrition education topics several DM food recommendations, 10.6% all about DM, and 7.3% management DM diet. “ But the figures 14.5% and 7.3% are not in Table 3 and I can’t understand what they mean by above description. It reworded to make it easier to understand.

We are realized that there is a mistake between table 2 and table 3. The individual nutrition education topics should be put in table 2. Including the data (percentage).

Therefore, we reworded to make it easier to understand.

5

3

Results

Age 58 years old.

Why did the authors use “58 years old” for stratification of patients by age? 60 or 65 years old may be usually used for such stratification.

The study used 58 years based on mean among 179 patients. Therefore, we used 58 years old.

5

4

Discussion

In the “Discussion” section, the authors should address limitation of their study. The number of patients, 179, is not necessarily large. No analyses is conducted for association between items in “Sociodemographic characteristics” *(Table 1) and those in “The need” (Table 3. Result from such association analyses may improve the quality of their work, which is expected in future studies.

As mentioned on the Discussion

9

Reviewer 2 Report

This is a cross-sectional study including 179 participants. Various nutritionists practices component were recorded and compared among 179 participants. Stratifications by the nutritionists’ practices component results were made regarding different nutrition education. They found that education factor had influence by referral of medical doctor.

This is an interesting study with some new findings in this area of research. The sample size of subjects is small for analysis. However, I nevertheless have the following comments that required to be addressed.

  1. The study design should be specified in title of this study. The authors should clarify this concern.
  2. The statistical methods used and described very poor. No statistical test used to analysis in this study. The authors should clarify this concern.
  3. How does the authors to determine the sample size of this study? Please use power analysis to statement adequate sample size in this study.
  4. For tables, I suggested to add tables with statistical test to provide clinical evidence for readers.
  5. We know the education as a risk factor in this study. How is the odds ratio (or relative risk) refer to males? The authors should clarify this concern.
  6. It seems the education as a common risk factor discussed in previous study. The authors should highlight novelty in this study. What do we newly learned from this study?
  7. Lastly, the authors only briefly discuss limitations, but didn’t acknowledging that the one of limitation include the cross-sectional design. They should elaborate on how the use of this design is subject to Incidence-Prevalence bias, also known as Neyman bias, and how that might influence their findings.

Author Response

Nutrients

MDPI, St. Anlage 66

CH-4052 Basel, Switzerland      

nutrients@mdpi.com

Telp:+41 61 683 77 34

 Fax: +41 61 302 89 18

Dear Editor of Nutrients, 

We would like to express tremendous gratitude for the email on 5th February 2022, and the opportunity to revise of manuscript ID- 1585719. We also thank the valuable comments from the reviewers that improved this final manuscript. 

The manuscript is revised based on the reviewer comments accordingly. We very much hope the revised manuscript is accepted for publication in Nutrients.

Thank you

Yours sincerely

Ice Yolanda Puri

  1. Reviewer-2

No

Item

Correction

Comments

Confirmation

Page

1

Title

The study design should be specified in title of this study. The authors

confirmed

1

2

Material and method

The statistical methods used and described very poor. No statistical test used to analysis in this study. The authors should clarify this concern.

In the study was not used statistical test due to determine T2DM patients’ perspective in the need the needs of individual nutrition education among people with T2DM

3

3

Material and method

How does the authors to determine the sample size of this study? Please use power analysis to statement adequate sample size in this study.

As mentioned at Material and Methods

3

4

Results

For table, I suggested to add tables with statistical test to provide clinical evidence for readers.

To have clinical evidence, may we put the results on the phase 3?

In the phase 3, we tested several evidence such as glycemic controls (HbA1c and fasting blood glucose), weight, Body Mass Index (BMI), Waist Circumference (WC), blood pressure, dietary intake (energy, carbohydrate, protein, fat, and fiber), knowledge-attitude-practice, and quality of life.

5

5

Results

We know the education as a risk factor in this study. How is the odds ratio (or relative risk) refer to males? The authors should clarify this concern.

A.   Phase 1

-      This is a phase 1 of the whole study which was separated into nutritionists’ perspective and T2DM patients’ perspective.

-      The study was a qualitative analysis to determine the needs of individual nutrition education among people with T2DM perspective.

-      Therefore, we thought that we cannot analyze using odd ratio (relative risk) refer to males. Majority of the subjects were female (76.5%), and the rest were males (23.5%).

B.   Phase 2

-      In the phase 2, we developed the Diabetes Nutrition Education (DNE) Module and materials based on the phase 1 results.

C.  Phase 3

-      Phase 3 was a cluster Randomized Controlled Trial to determine the effect of the DNE module for T2DM patients. The study was separated in intervention group (IG) and control group (CG). The individual nutrition education was done for 3 months and was followed by follow up for 3 months.

5

6

Discussion

It seems the education as a common risk factor discussed in previous study. The authors should highlight novelty in this study. What do we newly learned from this study?

-        We were heighted that the novelty of the study was T2DM patients attended the individual nutrition education based on referral of medical doctor.

-        Based on management practice at PHC by Indonesian Ministry of Health, individual nutrition education should receive nutrition counseling, besides anthropometry measurement, biomedical and clinical measurement.

-        Although, they were satisfied on individual nutrition education, unfortunately, T2DM patients were reluctant to attend the individual nutrition education.

-        The findings that can we were highlighted that T2DM patients need booklet as references of DM at home.

-        Based on our previous study among nutritionists’ perspective, T2DM patients were reluctant attending the individual nutrition education due to the tool kits were not interesting.

9

7

Discussion

Lastly, the authors only brief didn’t acknowledging that the one of limitation include the cross-sectional design. They should elaborate on how the use of this design is subject to Incidence-Prevalence bias, also known as Neyman bias, and how that might influence their findings.

The limitation of cross-sectional of our study were, we did not use quantitative data, therefore we cannot determine associate independent and dependent variable.

Besides, all of the T2DM patients were new diagnosed at least 6 months (incidence).

9

Round 2

Reviewer 2 Report

No further comments.